# Three Distinct Annotation Platforms Differ in Detection of Antimicrobial Resistance Genes in Long-Read, Short-Read, and Hybrid Sequences Derived from Total Genomic DNA or from Purified Plasmid DNA

**DOI:** 10.3390/antibiotics11101400

**Published:** 2022-10-12

**Authors:** Grazieli Maboni, Rodrigo de Paula Baptista, Joy Wireman, Isaac Framst, Anne O. Summers, Susan Sanchez

**Affiliations:** 1Athens Veterinary Diagnostic Laboratory, University of Georgia, Athens, GA 30602, USA; 2Department of Pathobiology, Ontario Veterinary College, University of Guelph, Guelph, ON N1G 2W1, Canada; 3Institute of Bioinformatics, University of Georgia, Athens, GA 30602, USA; 4Center for Tropical and Emerging Global Diseases, University of Georgia, Athens, GA 30602, USA; 5Department of Infectious Diseases, Houston Methodist Research Institute, Houston, TX 77030, USA; 6Department of Microbiology, University of Georgia, Athens, GA 30602, USA; 7Department of Infectious Diseases, University of Georgia, Athens, GA 30602, USA

**Keywords:** AMR prediction, plasmids, Nanopore sequencing, Illumina sequencing, whole genomes, WGS workflows

## Abstract

Recent advances and lower costs in rapid high-throughput sequencing have engendered hope that whole genome sequencing (WGS) might afford complete resistome characterization in bacterial isolates. WGS is particularly useful for the clinical characterization of fastidious and slow-growing bacteria. Despite its potential, several challenges should be addressed before adopting WGS to detect antimicrobial resistance (AMR) genes in the clinical laboratory. Here, with three distinct ESKAPE bacteria (*Enterococcus faecium*, *Staphylococcus aureus*, *Klebsiella pneumoniae*, *Acinetobacter baumannii*, *Pseudomonas aeruginosa*, and *Enterobacter* spp.), different approaches were compared to identify best practices for detecting AMR genes, including: total genomic DNA and plasmid DNA extractions, the solo assembly of Illumina short-reads and of Oxford Nanopore Technologies (ONT) long-reads, two hybrid assembly pipelines, and three *in silico* AMR databases. We also determined the susceptibility of each strain to 21 antimicrobials. We found that all AMR genes detected in pure plasmid DNA were also detectable in total genomic DNA, indicating that, at least in these three enterobacterial genera, the purification of plasmid DNA was not necessary to detect plasmid-borne AMR genes. Illumina short-reads used with ONT long-reads in either hybrid or polished assemblies of total genomic DNA enhanced the sensitivity and accuracy of AMR gene detection. Phenotypic susceptibility closely corresponded with genotypes identified by sequencing; however, the three AMR databases differed significantly in distinguishing mobile dedicated AMR genes from non-mobile chromosomal housekeeping genes in which rare spontaneous resistance mutations might occur. This study indicates that each method employed in a WGS workflow has an impact on the detection of AMR genes. A combination of short- and long-reads, followed by at least three different AMR databases, should be used for the consistent detection of such genes. Further, an additional step for plasmid DNA purification and sequencing may not be necessary. This study reveals the need for standardized biochemical and informatic procedures and database resources for consistent, reliable AMR genotyping to take full advantage of WGS in order to expedite patient treatment and track AMR genes within the hospital and community.

## 1. Introduction

Antimicrobial resistance is increasingly threatening global public health. Current routine drug susceptibility testing uses cultivation-based phenotyping with several commercial automated platforms [1]. Nonetheless, phenotyping may take three to four days for fast-growing bacteria, weeks for slow-growing bacteria [2,3], and considerably longer periods in non-automated laboratories. Interpretating susceptibility data is challenging since the lack of clinically relevant breakpoints for several pathogens [2,3] can delay decisions on adequate antimicrobial therapy.

Recent technical improvements and lower costs in rapid high-throughput DNA sequencing have engendered hope that whole genome sequencing (WGS) might afford the rapid detection of clinically relevant resistance genes. The consequent accurate prediction of resistance phenotypes could complement or even replace slower cultivation-based tests [4]. Genomic data have correctly predicted phenotypic resistance for some antimicrobial resistance (AMR) genes with established susceptibility breakpoints based on the minimum inhibitory concentration (MIC) of the corresponding antimicrobial [5,6,7,8]. However, WGS protocols for accurately detecting AMR genes are not yet standardized. This multistep process includes the specimen propagation in a suitable medium, DNA extraction, sequencing library preparation, generation of sequence ‘reads’, assembly of reads into chromosomes or plasmid sequences, and identification of antimicrobial resistance genes [9]. Since clinically relevant AMR genes are often carried on mobile genetic elements (transposons, integrons, and plasmids), the effective detection of these widely transmissible elements is essential. The purification of plasmid DNA is not as simple as that of total cellular (aka genomic) DNA, and there was concern that the latter might inefficiently recover large, low-copy-number plasmids [9,10,11], impairing the detection of plasmid-borne AMR genes. 

Sequence acquisition platforms and assembly pipelines deeply influence the fidelity of the final assembly, and thus, the accurate annotation of all genes encoded by chromosomes and plasmids [4]. The current industry-standard approach to full genome sequencing combines the highly accurate Illumina short-read sequencing with the newer, long-read platforms, such as Oxford Nanopore Technologies (ONT) or PacBio, which can better distinguish separate instances of repeated loci, the bete noire of genome assembly, especially in prokaryotes [12]. Nanopore’s long-read capacity is a real boon to sequencing plasmids, whose frequent repeated regions thwart the correct computational assembly of short-read data delivered by Illumina [6]. This advantage also applies to the assembly of bacterial whole genomes [12] because long-reads enable the correct structural resolution of complex genomic regions. The main drawback of using ONT sequencing alone is the higher error rate of raw sequence reads when compared to the more precise Illumina short-read technology [13]. Thus, combining short- and long-read sequencing has become the best practice for the sequencing of typically closed circular prokaryotic cell chromosomes (~4–5 Mb in enterobacteriaceae) and plasmids (ranging from 2–800 kb, also typically double-stranded closed circles) [12,14,15] to optimize the accurate annotation of all encoded genes, including AMR genes. 

Finally, identifying AMR genes requires reliable annotation databases of previously sequenced strains, including laboratory phenotypic data on their antimicrobial susceptibility. There are three frequently cited curated public databases dedicated only to AMR genes, each using different informatic strategies and data sources [4]: the Comprehensive Antibiotic Resistance Database (CARD) [14], ResFinder [15], and AMRFinder [16]. The outputs from these AMR databases often disagree with each other and with laboratory-based phenotyping [17]. This final all-important step is also very much in need of standardization. 

To devise a WGS best practice for the timely and accurate detection of relevant AMR genes in clinical isolates, the presence of such genes was investigated (i) in whole genome DNA or in purified plasmid DNA and (ii) by Nanopore long-read or Illumina short-read or a combination (hybrid). Then, for a given genome or plasmid sequence, we asked (iii) which public AMR reporting platform best identified clinically and epidemiologically relevant AMR genes and (iv) whether the genotype reported by an AMR database platform correlate with the laboratory-determined susceptibility phenotype of each strain.

## 2. Results

### 2.1. Concentration and Purity of Total Genomic DNA and Pure Plasmid DNA

The Genomic-tip 500/G kit protocol yielded the total genomic DNA with concentrations and purity, as described in Appendix A. The protocol was optimized for plasmid DNA extraction from liquid media to purify plasmids from bacterial colonies on an agar plate, eliminating 18 h overnight growth in liquid medium and yielding microgram amounts of pure plasmid DNA in 6 h (Appendix A; Figure 1). This was demonstrated by the positive control strain with the well-characterized 94 kb *E. coli* plasmid NR1 (control) [18]. Using gel electrophoresis, these plasmid-DNA preparations showed that the *E. ludwigii* had two plasmids of ~158.6 kb and ~7 kb and that the *K. pneumoniae* had four plasmids of ~232 kb, ~153 kb, ~71 kb, and ~6 kb (Figure 1).

### 2.2. Assembly and Assessment of Total Genomic DNA Preparations

The analytical pipeline is illustrated in Figure 2, and the total reads, the average read length, and the total base pairs detected are described in Appendix A. QUAST (Appendix A) reported chromosomes of the expected size for each strain using four distinct assembly approaches. Expected differences for long and short-read chemistries were high N50′s, low L50′s, and low N’s, indicating long-read Nanopore assemblies were less fragmented and more contiguous than the short-read Illumina assemblies. Total genomic coding sequences (CDS) annotated by Prokka in Nanopore-only assemblies (Appendix A) exceeded those of the GenBank reference sequences, but as expected, Nanopore-polished, Illumina-only, and Illumina-hybrid genomes agreed better with reference sequences (Appendix A).

### 2.3. Plasmid Sequences Assembled from Purified Plasmid DNA vs. Total Genomic DNA

For the *E. coli* (NR1) pure plasmid DNA sequences, Illumina-hybrid assembly generated a single closed contig of 94,308 bp corresponding to the previously determined 94 kb NR1 plasmid (control). In contrast, two or several linear contigs of different sizes resulted from the Nanopore-only, Illumina-only, and Nanopore-polished assemblies (Table 1). For the *E. ludwigii* pure plasmid DNA sequences, all four assembly approaches generated two linear contigs of ~130 kb and ~5 kb consistent with the ~158.6 kb and ~7 kb closed supercoiled plasmids seen using electrophoresis. For *K. pneumoniae* pure plasmid DNA sequences, four linear contigs were obtained with Nanopore-only and with Nanopore-polished assemblies, roughly corresponding to the supercoiled bands consistent with the ~232 kb, ~153 kb, ~71 kb, and ~6 kb bands visible in the gel. The smallest plasmid of *K. pneumoniae* was assembled into a single linear contig of 3–5 kb using all assembly approaches and was roughly similar to the ~6 kb supercoiled plasmid in the electrophoresis gel (Table 1).

The total genomic DNA preparations afforded the assembly of individual linear plasmids via the Nanopore-only, the Nanopore-Illumina, and hybrid methods for all bacterial strains (Table 1). As expected, Nanopore long-reads assembled into single linear contigs with sizes corresponding approximately to those observed in the purified plasmid preparations and on the corresponding gels, whereas Illumina short-reads assembled into several linear contigs of miscellaneous sizes.

The COPLA Taxonomic Classifier for plasmids [28] correctly identified the NR1 control as the incompatibility group (IncFII), mobility class (MOBF), and mating pair formation (type F) (Appendix A), placing it in the Plasmid Taxonomic Unit, PTU-FIIE. Taxonomic classifications of the two *E. ludwigii* and four *K. pneumoniae* plasmids were revealed by COPLA to belong to PTU-E3, PTU-HI1B, and PTU-E71III, respectively (Appendix A).

### 2.4. AMR Genes Detected in Total Genomic DNA vs. Plasmid DNA

An evaluation was made as to whether a plasmid extraction step must be incorporated in a sequencing workflow to obtain reliable identification of AMR genes. For that, AMR genes obtained from plasmid extraction only were compared against those from total genomic extractions. All AMR genes detected in the plasmid extractions were also present in the total genomic DNA extractions of all bacterial strains investigated (Table 2). Further, more AMR genes were detected in the total genomic data than in plasmid-only data, independent of the assembly approach applied (see Appendix A for a complete list of genes). In *K. pneumoniae*, genes expected to be detected in plasmids were only found in the total genomic DNA (i.e., *aadA1* and *qacEdelta1*) (Table 2).

### 2.5. Phenotypic vs. Genotypic Antimicrobial Susceptibility

The MIC interpretation of Vitek-2 and Sensititre systems were consistent, but two discrepant results were observed in *E. ludwigii* and *K. pneumoniae*. For instance, *E. cloacae*’s MIC to chloramphenicol indicated intermediate resistance by VITEK-2, while it was susceptible by Sensititre. Similarly, the MIC of *K. pneumoniae* to amoxicillin/clavulanic acid was susceptible by Vitek-2 but resistant by Sensititre (Table 3). 

The ability of the WGS data to correctly identify AMR genes associated with a resistant phenotype was subsequently evaluated. For that, phenotypic resistance was compared with the predicted phenotype based on the presence of AMR genes. WGS data elucidate some of the expected discrepancies observed between Vitek-2 and Sensititre; for instance, the *oqxA* and *oqxB* genes conferring resistance to chloramphenicol were detected in *E. ludwigii* assemblies, and the *blaSHV* gene conferring resistance to beta-lactams was detected in *K. pneumoniae* assemblies (Table 4). As proof of the efficacy of the approaches applied, AMR genes corresponded to the phenotypes of resistance to tetracyclines and chloramphenicol within the total genomic DNA, as well as plasmid DNA, preparations of the NR1 plasmid (control) as previously characterized (Table 4). However, these results varied depending on the AMR database applied, as described in Section 2.6 and Table 4.

### 2.6. Comparison of AMR Genes Databases

To evaluate the efficiency with which AMR genes could be detected from different publicly available databases, the gene symbol output obtained from each database in total-genomic preparations were compared. Most genes called by AMRFinder and ResFinder were identical (*n* = 7), with 14 genes in total being called by AMRFinder and 9 genes in total called by ResFinder, while CARD called 88 AMR genes among the three bacterial strains (Figure 3; Appendix A). Besides having great database resources, these results highlight the stringent search of AMRFinder and ResFinder, which contributes to avoid false positives. In contrast, CARD uses less stringent cut-off thresholds, increasing false positives and overcalling resistance.

We then compared AMR-gene outputs of AMRFinder, ResFinder, and CARD with phenotypic resistance results. AMRFinder detected the *blaEC* gene in the *E. coli* strain, and *oqxA*/*oqxB* genes in *E. ludwigii*, which are consistent with their phenotypic resistance to beta-lactams and chloramphenicol (Table 4, Figure 3). In contrast, ResFinder missed these two categories of phenotypic resistance, and CARD missed one category (chloramphenicol in *E. ludwigii*) (Table 4).

AMRFinder additionally screened for biocide and metal resistance genes, identifying the *qacDeltaE1* gene (quaternary ammonium) in *K. pneumoniae* and *E. coli* (Figure 3), but it missed well-characterized mercury-resistance conferring genes, such as the *merA*, in the *E. coli* plasmid (control) [18].

### 2.7. Comparision of Sequencing Platforms and Assembly Approaches for Detection of AMR Genes

Illumina-only and hybrid or Nanopore-polished approaches allowed for the more accurate detection of AMR genes in the control strain (*E. coli*) than in the Nanopore-only assemblies (Table 5). For instance, the *sul1* and *tet(B)* genes were not detected in Nanopore-only assemblies of *E. coli*, therefore, failing to predict resistance to chloramphenicol and tetracyclines, to which *E. coli* was shown to be resistant in the phenotypic in vitro testing (Table 4 and Table 5). Similarly, genes conferring resistance to chloramphenicol (*oqxA* and *oqxB*) were not detected in Nanopore-only assemblies of *E. ludwigii*, therefore, failing to predict phenotypic resistance to chloramphenicol (Table 4 and Table 5). This confirms that the accuracy of the base-calling and assembly methods influences the identification and order of the bases in the output of any sequencing platform.

For the two test strains, *E. ludwigii* and *K. pneumoniae*, AMR genes expected to confer phenotypic resistance to chloramphenicol and beta-lactams (Table 4) were not detected in plasmids sequenced by Nanopore and assembled by Flye (Nanopore-only and Nanopore-polished approaches). Therefore, further investigation was focused on identifying whether the issue was related to the type of plasmid assembler employed. We found that assembling plasmids with different long-read tools (i.e., Flye versus Canu) affected the detection of AMR genes (Appendix A). Genes expected to be detected in *E. ludwigii* plasmids, such as *fosA* and *blaACT,* were identified in assemblies from Canu but not from Flye (Appendix A). However, in *K. pneumoniae*, either Flye or Canu failed to improve the detection of *blaSHV-40* and *fosAgenes* in plasmid-only assemblies (Appendix A).

## 3. Discussion

While there has been a considerable progress in the cost and availability of WGS, integrating these technologies into routine clinical microbiology is challenging. To our knowledge, this is a pioneering study that critically assesses each step in AMR-gene detection in WGS. This study motivates further investigations in identifying the best practices of DNA extraction, sequencing platforms, assembly methodologies, and publicly available databases for clinically relevant bacteria, e.g., the ESKAPE list (*Enterococcus faecium*, *Staphylococcus aureus*, *Klebsiella pneumoniae*, *Acinetobacter baumannii*, *Pseudomonas aeruginosa*, and *Enterobacter* spp.). The results from this study have extensive application, especially since the evaluated methods are also frequently used for other bacterial pathogens.

Most AMR genes are located in mobile elements, and DNA extraction methods may impair the extraction of plasmids [9,11]; therefore, it was investigated whether a plasmid extraction step needed to be incorporated in a sequencing workflow to obtain a consistent identification of AMR genes. In this study, a separate plasmid extraction step did not yield better results since all AMR genes detected in the plasmid-only assemblies were also present in the total genomic DNA assemblies and corresponded in size and abundance to plasmids as observed by gel electrophoresis. The literature evaluating the potential impact of DNA extraction workflows on subsequent AMR detection is limited. Salting-out kits have exhibited difficulty extracting small plasmids from *K. pneumoniae* [9] and presented impaired plasmid extraction performance in *E. coli* [11]. However, it did not influence the data quality and also inferred phylogenetic relationships of Illumina-generated *E. coli* WGS [11,29]. A solid-phase (anion-exchange) total genomic DNA kit, Genomic-tip 20/G-Qiagen, was employed in this study and has been demonstrated to rapidly and inexpensively provide sufficient sequencing-quality plasmid DNA [11]. Plasmid extractions can be cumbersome and not adaptable in a clinical diagnostic setting; therefore, an existing commercial extraction kit for purifying plasmids directly from bacterial colonies harvested from an agar plate was adapted in this study, resulting in a turnaround time of 6 h and yielding a high concentration and quality of plasmid DNA, as suggested by ONT’s DNA-quality recommendations. However, despite the faster turnaround time, the extraction protocol, followed by the sequencing and assembling steps, is still cumbersome and may not add any additional information, as demonstrated in this study.

Further, the ability of WGS approaches to identify AMR genes associated with an experimental phenotype was evaluated. Using MICs provided by automated susceptibility testing as a gold standard, the presence of AMR genes was found to be a good predictor of resistance phenotypes but not a good predictor of susceptibility phenotypes. As MICs were considered the gold standard, two different methodologies were used to ensure MICs since discrepancies observed between phenotypic results and the expected genomic outcome are often caused by incorrect susceptibility testing [4]. Regarding resistance prediction, we showed that all categories of phenotypic resistance could be associated with AMR genes based on AMRFinder database results. Recent studies demonstrated the power to predict AMR from genomic data in clinical isolates of *E. coli*, *K. pneumoniae*, and *E. cloacae* [6,8]. Although this is promising, we do not suggest that WGS can replace phenotypic susceptibility testing, but rather, it serves as a complementary method that is particularly useful for testing fastidious and slow-growing bacteria [30]. It can also be useful in cases of discrepant MIC results between different susceptibility methods as observed in *K. pneumoniae*, where the detection of the *blaSHV* gene could clarify the interpretation of discrepant MICs for amoxicillin/clavulanic acid.

Significant variation was observed among the three AMR databases in reporting bona fide mobile antibiotic resistances versus chromosomal housekeeping genes in which rare spontaneous resistance mutations could occur. The presence of such housekeeping genes was associated with susceptible MIC values, therefore, failing to predict susceptibility. Such results may have clinical implications considering that the detection of chromosomal housekeeping genes, overcalled as “AMR genes”, could mislead clinicians to believe that the bacterium is resistant where it is susceptible. CARD results consisted mainly of chromosomal genes that confer the transient up-regulation of efflux pumps or redox stress defense and may barely confer clinically relevant resistance. As CARD does not include a mobile genes database [14], many of the transient genes from chromosomal DNA were detected instead of AMR genes from mobile elements, and many of the listed genes only have the potential to become resistant without intrinsically conferring high resistance levels. The limited stringency of CARD may impair the practicality of gene output interpretation in a clinical context. AMRFinder showed to be a valuable resource for acquired AMR genes, however, AMRFinder screening of biocide and metal resistance was in complete given that it missed the well-characterized mercury-resistance conferring gene in the *E. coli* plasmid (control) [18]. Based on these data, AMRFinder and ResFinder provided easy output interpretation, a low number of overcalled “AMR genes”, and a good prediction of resistance phenotypes. Therefore, the adoption of at least two in silico databases in a clinical setting should be used to compare their outcomes to precisely identify AMR genes.

This study demonstrated that the methods of choice may significantly influence the detection of AMR genes. Further, this study demonstrated that plasmid DNA could be extracted and sequenced as well as chromosomal DNA using this study’s total genomic DNA protocol, and possibly even better, because many AMR genes are chromosomally borne. As plasmid extraction, sequencing, and assembling protocols were cumbersome and did not provide any additional information besides the findings from the total genomic DNA extraction, it is suggested that a plasmid DNA extraction/sequencing step may not be essential to obtain a complete list of AMR genes. Nanopore-only sequences missed AMR genes and failed to predict *E. coli* and *E. ludwigii* phenotypic resistance. The Illumina-only approach was as accurate as the hybrid assemblies, but it was still slow and expensive if used for critical care. The pace at which microfluidics and nanochemistries are addressing the AMR-detection problem may soon remedy the challenges of cost and accuracy of data acquisition. It will be particularly useful for the AMR genotyping of fastidious and slow-growing bacteria in a clinical context. For the moment, creative deployment of a mix of existing rapid data acquisition with real-time in-house data collection, processing, and modeling will improve the ability of clinical laboratories to handle challenging cases rapidly and to expedite patient treatment and AMR tracking.

## 4. Materials and Methods

### 4.1. Bacterial Strains

Two clinical bacterial strains, *Enterobacter ludwigii* (LST1391B) and *Klebsiella pneumoniae* (LST1504-C2) were isolated at University of Georgia from MacConkey agar plates inoculated with fecal swabs from two unrelated hospital patients provided by the Stuart Levy lab at Tufts University Medical Center in Boston, MA, and were cryopreserved. As a control, we used the cryopreserved standard *Escherichia coli* laboratory strain (DU1040) carrying the extensively characterized 94-kb conjugative IncFII plasmid, NR1 [18]. The cryopreserved strains were revived by streaking on 5% sheep blood agar (Remel, San Diego, CA, USA) and incubated for 24 h at 35 °C with 5% CO_2_.

### 4.2. Antimicrobial Susceptibility Testing

The minimum inhibitory concentration (MIC) of antimicrobials was determined using two systems, Vitek-2 (bioMérieux, Marcy l’Etoile, France) and Trek Sensititre (Trek Diagnostic Systems, Cleveland, OH, USA). For the Vitek-2 testing, three different MIC cards (GN-98, GN-69, and GN-82) were run according to the manufacturer’s instructions (Vitek-2, bioMérieux). For the Trek Sensititre testing, both GN4F and COMPGN1F Gram-negative microplates were run according to the manufacturer’s instructions (Trek Diagnostic Systems). Twenty-one antimicrobials representing eight chemical classes of drugs were tested: lactams and lactamase inhibitors (ampicillin, amoxicillin/clavulanic acid, piperacillin/tazobactam, cefalexin, ceftriaxone, cefazolin, cefepime, ceftazidime); fluoroquinolones (ciprofloxacin, levofloxacin, enrofloxacin); aminoglycosides (gentamicin, amikacin); tetracyclines (doxycycline, tetracycline); antifolates (trimethoprim/sulfamethoxazole); carbapenems (ertapenem, imipenem, meropenem); phenicol (chloramphenicol); and nitrofurans (nitrofurantoin). The resulting MIC value was assigned to clinical categories of susceptible or resistant according to the Clinical & Laboratory Standards Institute (CLSI M-100) [2] and the European Committee on Antimicrobial Susceptibility Testing (EUCAST) [3]. MIC results in “intermediate” or “resistant” ranges were both assigned as “resistant”.

### 4.3. Extraction of Total Genomic DNA and Pure Plasmid DNA

To expedite the plasmid purification, we used fresh colonies from non-selective agar rather than a liquid broth culture. Specifically, cryopreserved bacterial cells were streaked for colony isolation on Luria-Bertani agar without antibiotics (Remel) and incubated for 18 h at 35 °C with 5% CO_2_. All growth on agar was scraped from the plate surface, transferred into 500 mL of Luria-Bertani broth (Remel), and incubated at 35 °C, at 120 rpm for 3 to 4 h until approximately mid-exponential phase (OD600 nm = 0.6). Then the entire culture was centrifuged at 6000 rpm for 15 min at 4 °C; the supernatant-spent medium was discarded, and plasmids were extracted from the cell pellet using the Qiagen Large Construct Kit. According to the manufacturer’s instructions, total DNA extraction was performed on this suspension of freshly grown bacterial colonies using the Genomic-tip 500/G kit (Qiagen, Hilden, Germany). Plasmids were separated by standard horizontal electrophoresis in 0.5% SeaKem Gold (Lonza) agarose gel in Tris-acetate buffer (39.6 mMTris/8.2 mM sodium acetate, pH 8, at room temperature, along with supercoiled DNA standards (BacTracker, Epicentre Biotechnologies, Madison, WI, USA). Gels were run at 35 volts until the running dye reached the lower edge of the gel, approximately 18 h. The gels were then stained with SyberGreen I (Sigma Aldrich) and imaged as previously described [18]. The plasmid molecular weight was estimated according to a semi-log polynomial fit of the migration distances of standard supercoiled plasmid DNA bands of known molecular mass.

The concentration of total genomic or plasmid DNA preparations was quantified by a Qubit 2.0 fluorometer using a double-stranded DNA assay kit. Purity was assessed by NanoDrop Spectrophotometer, according to the manufacturer’s instructions (Thermo Scientific, Waltham, MA, USA). DNA preparations were stored at −20 °C until sequencing.

### 4.4. Whole-Genome and Plasmid Sequencing

MinION libraries were prepared from 400 ng of pure plasmid or total genomic DNA using the SQK-RBK004 Rapid Barcoding Kit and sequenced using the FLOW-MIN 106 (R9.4 SpotOn) flow cell according to instructions from ONT. The total genomic DNA and pure plasmid DNA of each strain were barcoded separately. ONT’s MinKNOW software (version v18.03.1) collected raw electronic data as Fast5 read files, and bases were called using ONT’s EPI2ME software. Initial real-time workflows “What Is in My Pot?” (WIMP) were used to confirm bacterial biotypes based on 16S rDNA. Sequence data were collected for 24 h.

Illumina paired-end libraries were prepared from total genomic DNA or pure plasmid DNA using a Nextera DNA Flex library prep kit on an Illumina iSeq 100 instrument and sequenced with 150 bp paired reads according to the manufacturer’s instructions. Quality control of library preparation was performed using the QIAxcel Advanced Systems (Qiagen, Hilden, Germany). Nanopore and Illumina sequencing was performed at the Athens Veterinary Diagnostic Laboratory, University of Georgia, Athens, GA, USA.

### 4.5. Contig Assembly and Data Analysis

All raw reads were submitted to a metagenomics pipeline to detect species or potential contamination using Metaphlan2 v2.7.8 before the de novo assembly [20]. Each genome or plasmid was assembled using four different strategies: (i) Nanopore long-reads, named in this study as the Nanopore-only approach; (ii) Illumina short-reads, named in this study as the Illumina-only approach; (iii) simultaneous assembly of Illumina reads and Nanopore reads, named as the Hybrid approach; (iv) and the Flye-assembled Nanopore reads post-hoc matched with Illumina reads, named as the Nanopore-polished approach. Assembly was guided by the average published chromosome size for the respective bacterial genus available in GenBank. The assembly of pure plasmid sequences was guided by the estimated molecular size of the plasmids observed in agarose gel electrophoresis (Figure 1).

For the Nanopore-only assembly, barcoded sequencing reads were demultiplexed using Porechop v0.2.4 (https://github.com/rrwick/Porechop accessed on 11 September 2022) and assembled using Flye v2.6 [21]. Potential base-call errors in the assembly were verified by Racon v.1.4.7 [25], which generates a genomic consensus with better quality than the output generated through assembly methods using the alignment coverage of the contig blocks. An additional correction step was made in the previous Nanopore-only assembly using Illumina reads for the Nanopore-Illumina polished approach. This correction was done by combining two alignment iterations using BWA v0.7.17 [31] with the MEM function and SMALT v0.7.4 (https://www.sanger.ac.uk/tool/smalt-0/ accessed on 11 September 2022); the output alignment was submitted to the polishing tool Pilon v.1.23 [24].

Besides Flye v.2.6, the assembly of plasmid raw reads generated by ONT was also performed using Canu v.2.2 [23]. For the Nanopore-only plasmid assembly with Canu, trimmed reads from Porechop were input to Canu v.2.2 using the default options for ONT reads. For Nanopore-polished plasmid assemblies, BWA-MEM v0.7.17 [31] was used to generate alignment overlaps between the ONT plasmid draft assembly and the Illumina reads generated from the plasmid prep. The alignment was then parsed using Samtools v1.12 [28] and used for base-call polishing in two rounds of Pilon v1.23 [32].

All Illumina paired-end read sequences generated were quality checked using FastQC and trimmed by Trimomatic v.0.36 [31] to remove sequencing adapters and reads with Phred scores < 30. The de novo assembly of Illumina reads was performed using Spades v3.12 [22] and polished by Pilon v.1.23 using the same Illumina alignment protocol applied for the Nanopore-Illumina polished approach described above. For the Illumina-hybrid approach, both Illumina and Nanopore reads were submitted to a de novo assembly using hybridSPAdes [33] incorporated in Spades v3.12 and were polished by Pilon. This approach prioritizes the contig formation by using de Bruijn graphs with the Illumina short-reads and then mapping long-reads in the edges of the assembly graph to increase contiguity and generate longer scaffolds. PlasmidSPADES, also available in Spades v3.12, was used to optimize plasmid-contig assembly from the Illumina data [26]. Taxonomic classification of plasmids was performed using COPLA v1.0 [28]. Assembly statistics were assessed using QUAST v5.0.2 [34]. Annotations of chromosomes and plasmids were performed using Prokka v1.13 [27].

Our bioinformatic pipeline shell script with dependencies, installation instructions, and usage instructions is available at https://github.com/iframst/HybridAMRgenotyping (accessed on 11 September 2022).

### 4.6. Identification of AMR Genes in Whole Genome or Pure Plasmid Sequences

AMR genes were identified in the whole genome and plasmid sequences using three databases (i) ResFinder for acquired AMR genes, with default settings of 90% nucleotide similarity and a 60% minimum length [19]; (ii) Comprehensive Antibiotic Resistance Database (CARD) with select criteria as follows: perfect and strict hits only, excluded nudging of ≥95% identity loose hits to strict, and high quality/coverage sequences [14]; and (iii) AMRFinder from the National Center for Biotechnology Information (NCBI) with minimum BLAST identity cut-off of >90% and >50% alignment coverage; and organism search for optimal analysis of *E. coli* and *K. pneumoniae*. Venn diagrams were performed using Venny (version 2.1.0) [35].

## Figures and Tables

**Figure 1 antibiotics-11-01400-f001:**
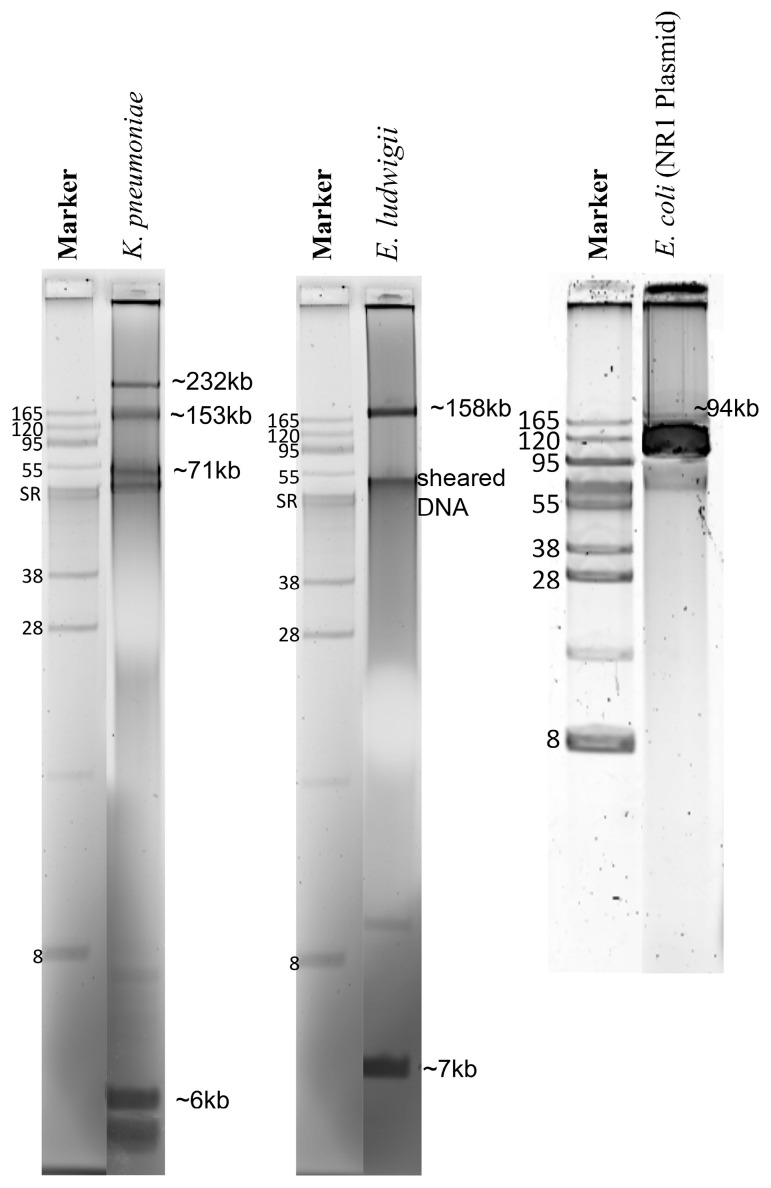
Agarose electrophoresis of plasmids from three enterobacteria sequenced in this study. The gel used is 0.5% SeaKem Gold agarose stained with SyberGreen. Plasmid molecular weight (kilobases = kb) was estimated using semi-log plotting based on DNA-band sizes observed in the agarose gel electrophoresis. Strains: *Klebsiella pneumoniae*—LST1504-C2, *Enterobacter ludwigii*—LST1391B, and *Escherichia coli*—DU1040 (NR1) plasmid control.

**Figure 2 antibiotics-11-01400-f002:**
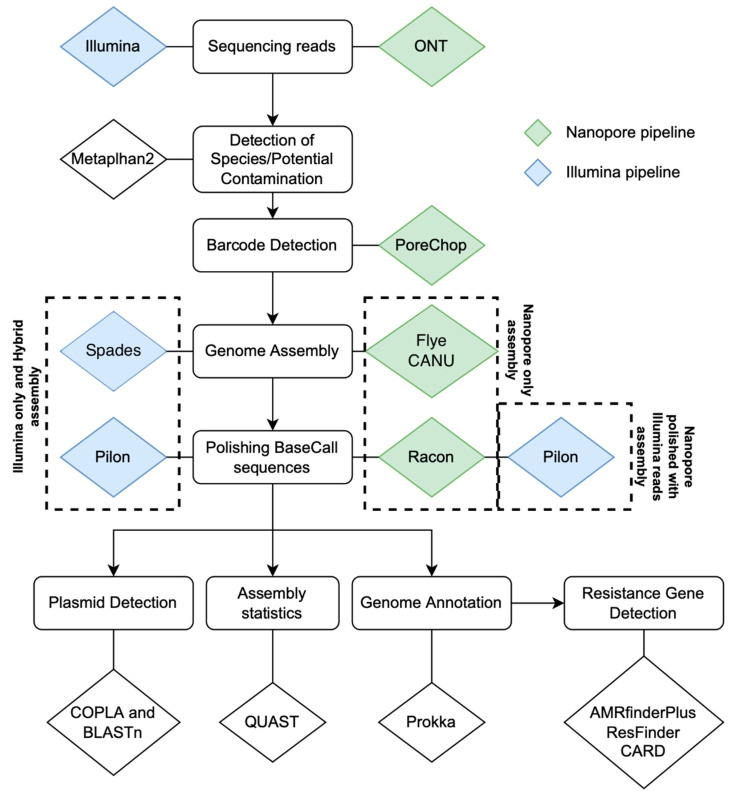
The bioinformatics pipeline used for the total genomic and plasmid DNA sequencing analysis. The total genomic DNA and plasmid-only DNA were sequenced using two sequencing chemistries (Illumina and Oxford Nanopore Technologies (ONT)). Reads were assembled using four different approaches: Nanopore-only: MinION reads were assembled with Flye, without error correction by Illumina reads; Nanopore-polished: MinION reads were assembled with Flye and polished with Illumina reads; and Illumina-only: Illumina reads were assembled with SPADES, without error correction with Nanopore reads. Illumina-hybrid: assembly of Illumina reads and Nanopore reads with hybridSPADES and polished with Illumina reads. Blue triangles: Illumina. Green triangles: ONT. AMR genes were identified using three different databases: AMRfinder database [16], Resfinder database [19], and CARD [14]. Metaphlan2 [20], Porechop (https://github.com/rrwick/Porechop accessed on 11 September 2022), Flye [21], Spades [22], Canu [23], Pilon [24], Racon [25], PlasmidSPAdes [26], Prokka [27], Copla [28].

**Figure 3 antibiotics-11-01400-f003:**
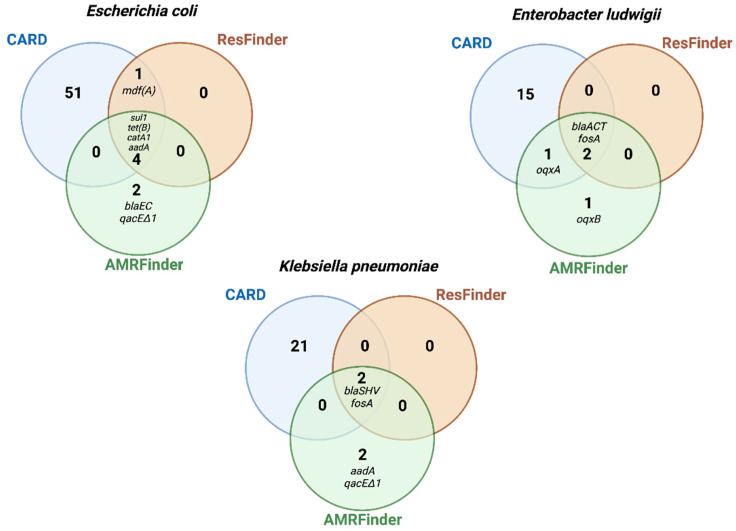
Antimicrobial resistance (AMR) genes with matches in three databases: AMRFinder, ResFinder, and Comprehensive Antibiotic Resistance Database (CARD). Venn diagrams are based on total genomic DNA sequences (Illumina-hybrid approach), and sizes do not reflect the number of AMR genes. See Appendix A for the description of genes listed by CARD.

**Table 1 antibiotics-11-01400-t001:** Plasmid sequences inferred from the Nanopore and/or Illumina sequencing assemblies of pure plasmid DNA or of total genomic DNA.

Strain	Plasmid Size (kb) ^a^	Nanopore- Only ^b^ Contig (bp)	Nanopore- Polished ^c^ Contig (bp)	Illumina-Only ^d^ Contig (bp)	Illumina-Hybrid ^e^ Contig (bp)
**Pure Plasmid DNA**					
*E. coli* DU1040 (NR1)	94.289	Several contigs ^f^	93,868	Several contigs	94,308
*K. pneumoniae*	232	313,736	313,014	Several contigs	Several
LST1504-C2	153	265,010	265,397	Several contigs	Several
	71	58,149	58,214	Several contigs	Several
	6	3380	3378	3003	6261
*E. ludwigii* LST1391B	158.6	133,397	133,378	130,070	130,070
	7	5187	5152	5283	5152
**Total Genomic DNA**					
*E. coli* DU1040 (NR1)	94.289	94,296	94,410	Several contigs	93,656
*K. pneumoniae*	232	313,645	312,971	Several contigs	314,384
LST1504-C2	153	264,823	264,679	Several contigs	Several
	71	58,157	58,118	Several contigs	Several
	6	3350	3326	2930	2930
*E. ludwigii* LST1391B	158.6	133,990	133,333	130,070	130,070
	7	6174	6174	Several contigs	5033

^a^ Plasmid sizes were roughly estimated using standard electrophoresis. ^b^ Nanopore-only: MinION reads were assembled using Flye, without error correction of lllumina reads. ^c^ Nanopore-polished: MinION reads were assembled using Flye and polished with Illumina reads. ^d^ Illumina-only: Illumina reads were assembled using SPADES, without error correction with Nanopore reads. ^e^ Illumina-hybrid: assembly of Illumina reads and Nanopore reads with hybridSPADES, polished with Illumina reads. ^f^ Several: two or more contigs of different lengths matched with the control or with a plasmid sequence from GenBank.

**Table 2 antibiotics-11-01400-t002:** Antimicrobial resistance (AMR) genes detected in the total genomic DNA vs. in pure plasmid DNA preparations.

ResFinder		AMRFinder	
Total Genomic	Pure Plasmid	Total Genomic	Pure Plasmid
*Escherichia coli* DU1040 (NR1)			
* **aadA1** *	* **aadA1** *	* **aadA1** *	* **aadA1** *
* **catA1** *	* **catA1** *	* **catA1** *	* **catA1** *
* **sul1** *	* **sul1** *	* **sul1** *	* **sul1** *
* **tet(B)** *	* **tet(B)** *	* **tet(B)** *	* **tet(B)** *
*mdfA*		* **qacEdelta1** *	* **qacEdelta1** *
		*blaEC*	
*Enterobacter ludwigii* LST1391B			
* **blaACT-12** *	* **blaACT-12** *	* **blaACT** *	* **blaACT** *
* **fosA2** *	* **fosA2** *	* **fosA2** *	* **fosA2** *
		* **oqxA** *	* **oqxA** *
		* **oqxB** *	* **oqxB** *
*Klebsiella pneumoniae* LST1504-C2			
*blaSHV-40*	Not identified ^a^	*blaSHV*	Not identified ^b^
*fosA*		*fosA*	
		*aadA1*	
		*qacEdelta1*	

^a^ Results based on Illumina-hybrid assemblies, which exhibited more AMR genes. See Methods and Table 1 footnote for the details of assembly pipelines. ^b^ Not identified: antimicrobial resistance genes were not detected. Bold: genes detected in plasmids that were observed in both the pure plasmid and total genomic DNA preparations. Aminoglycosides: *aadA1*. Beta-lactams: *blaACT*, *blaSHV*, and *blaEC*. Phenicol: *catA1*. Fosfomycin: *fosA2*. Macrolide, tetracycline, phenicol (broad spectrum): *mdf(A)*. Tetracycline *tet(B)*. Sulphonamide: *sul1*. Phenicol/Quinolone: *oqxA*, *oqxB*. Quaternary ammonium: *qacEdelta1.*

**Table 3 antibiotics-11-01400-t003:** Minimum inhibitory concentration (MIC) of antimicrobials for the standard and two test strains using two methods, Sensititre and Vitek-2.

Antimicrobial	*E. coli* DU1040 (NR1)		*E. ludwigii* LST1391B		*K. pneumoniae* LST1504-C2	
	MIC Vitek-2	MIC Sensititre	MIC Vitek-2	MIC Sensititre	MIC Vitek-2	MIC Sensititre
Ampicillin	8 (S)	≤8 (S)	NI	**>16 (R)**	**16 (R)**	**>16 (R)**
Amox./Clavulanic Acid	4 (S)	4 (S)	**≥32 (R)**	**2 (R)**	≤2 S	**>8 (R)**
Piperacillin/Tazobactam	≤4 (S)	≤8 (S)	≤4 (S)	≤8 (S)	≤4 (S)	≤8 (S)
Cephalexin	**16 (R)**	**16 (R)**	**≥64 (R)**	**8 (R)**	**≥16 (R)**	**>16 (R)**
Ceftriaxone	≤1 (S)	≤0.5 (S)	≤1 (S)	≤0.5 (S)	≤1 (S)	≤0.5 (S)
Cefazolin	≤4 (S)	4 (S)	**≥64 (R)**	**2 (R)**	**32 (R)**	**>32 (R)**
Cefepime	≤1 (S)	≤4 (S)	≤1 (S)	≤4 (S)	≤1 (S)	≤4 (S)
Ceftazidime	≤1 (S)	≤1 (S)	≤1 (S)	≤1 (S)	≤1 (S)	≤1 (S)
Ciprofloxacin	≤0.25 (S)	≤0.5 (S)	≤0.25 (S)	≤0.5 (S)	≤0.25 (S)	≤0.5 (S)
Levofloxacin	≤0.12 (S)	≤1 (S)	≤0.12 (S)	≤1 (S)	≤0.12 (S)	≤1 (S)
Enrofloxacin	0.5 (S)	0.25 (S)	≤0.12 (S)	≤0.125 (S)	≤0.12 (S)	≤0.125 (S)
Gentamicin	≤1 (S)	≤0.5 (S)	≤1 (S)	≤0.5 (S)	≤1 (S)	≤1 (S)
Amikacin	≤2 (S)	≤4 (S)	≤2 (S)	≤4 (S)	≤2 (S)	≤4 (S)
Doxycycline	**≥16 (R)**	**≥8 (R)**	4 (S)	2 (S)	1 (S)	4 (S)
Tetracycline	**≥16 (R)**	**≥16 (R)**	≤4 (S)	≤4 (S)	4 (S)	≤4 (S)
Chloramphenicol	**≥64 (R)**	**≥32 (R)**	**16 (I)**	4 (S)	≤2 (S)	2 (S)
Nitrofurantoin	≤16 (S)	≤32 (S)	32 (S)	≤32 (S)	≤16 (S)	≤32 (S)
Trim./Sulfamethoxazole	≤2 (S)	≤0.5 (S)	≤20 (S)	≤0.5 (S)	≤20 (S)	≤0.5 (S)
Ertapenem	≤0.5 (S)	≤0.25 (S)	≤0.5 (S)	≤0.25 (S)	≤5 (S)	≤0.25 (S)
Imipenem	≤0.25 (S)	≤0.5 (S)	0.5 (S)	≤0.5 (S)	≤0.25 (S)	≤0.5 (S)
Meropenem	≤0.25 (S)	≤0.5 (S)	≤0.25 (S)	≤0.5 (S)	≤0.25 (S)	≤0.5 (S)

In bold, R: resistant; I: intermediate resistance. NI: no interpretation provided by Vitek-2 and Trek Sensititre systems. Interpretations were based on the Clinical & Laboratory Standards Institute (CLSI M-100) and the European Committee on Antimicrobial Susceptibility Testing (EUCAST, 2019) recommendations.

**Table 4 antibiotics-11-01400-t004:** Correspondence of experimentally determined resistance phenotypes versus AMR genes detected in total genomic DNA.

Strain	Resistance Phenotype ^b^	Antibiotic Class	Genes Detected by		
			ResFinder	AMRFinder	CARD
*E. coli* DU1040 (NR1)	Cefalexin	Beta-lactam	ND	*blaEC*	*Amp-C* and *Amp-H Beta-lactamases*
	Doxycycline	Tetracycline	*tetB*, *mdf(A)*	*tetB*	*tetB*
	Tetracycline	Tetracycline	*tetB*, *mdf(A)*	*tetB*	*tetB*, *mdf(A)*
	Chloramphenicol	Phenicol	*catA1*, *mdf(A)*	*catA1*	*catA1*, *mdf(A)*
*E. ludwigii*^a^ LST1391B	Ampicillin	Beta-lactam	*blaACT-12*	*blaACT*	*blaACT-12* and *Amp-H Beta-lactamases*
	Amoxicillin/clavulanic acid	Beta-lactam	*blaACT-12*	*blaACT*	*blaACT-12* and *Amp-H Beta-lactamases*
	Cefazolin	Beta-lactam	*blaACT-12*	*blaACT*	*blaACT-12* and *Amp-H Beta-lactamases*
	Cefalexin	Beta-lactam	*blaACT-12*	*blaACT-12*	*blaACT-12* and *Amp-H Beta-lactamases*
	Chloramphenicol	Phenicol	ND	*oqxA*, *oqxB*	ND
*K. pneumoniae* ^a^ LST1504-C2	Ampicillin	Beta-lactam	*blaSHV-40*	*blaSHV*	*blaSHV-40* and *Amp-H Beta-lactamases*
	Amoxicillin/clavulanic acid	Beta-lactam	*blaSHV-40*	*blaSHV*	*blaSHV-40* and *Amp-H Beta-lactamases*
	Cefazolin	Beta-lactam	*blaSHV-40*	*blaSHV*	*blaSHV-40* and *Amp-H Beta-lactamases*
	Cephalexin	Beta-lactam	*blaSHV-40*	*blaSHV*	*blaSHV-40* and *Amp-H Beta-lactamases*

AMR Annotation databases: AMRFinder, ResFinder and CARD. Results are based on Illumina-hybrid, Nanopore-polished, and Illumina-only assemblies (see Methods section and the footnote of Table 1 for the details of assembly pipelines). ^a^ The *Enterobacter cloacae* complex is intrinsically resistant to ampicillin, amoxicillin/clavulanic acid, and cefazolin. *K. pneumoniae* is intrinsically resistant to ampicillin. ^b^ No resistance genes corresponding to this resistance phenotype were detected in the Nanopore-only assembly, but AMR genes were detected in Nanopore-hybrid, Illumina-only, and Illumina-hybrid using the AMRFinder database. ND: not detected.

**Table 5 antibiotics-11-01400-t005:** Comparison of two sequencing chemistries, four assembly pipelines, and two databases for the detection of antimicrobial resistance (AMR) genes in the total genomic DNA.

	ASSEMBLY ^a^ >	Nanopore-Only	Nanopore-Polished	Illumina-Only	Illumina-Hybrid	Nanopore-Only	Nanopore-Polished	Illumina-Only	Illumina-Hybrid
	DATABASE>	ResFinder database				AMRFinder database			
AMR GENE ^b^	BACTERIUM								
	*E. coli* DU1040 (control, NR1)								
*aadA1*		+	+	+	+	+	+	+	+
*blaEC*		ND	ND	ND	ND	+	+	+	+
*catA1*		+	+	+	+	+	+	+	+
*mdf(A)*		+	+	+	+	ND	ND	ND	ND
*qacEΔ1*		NA	NA	NA	NA	+	+	+	+
*sul1*		+	+	+	+	ND	+	+	+
*tet(B)*		+	+	+	+	ND	+	+	+
	*E. ludwigii* LST1391B								
*blaACT-12*		+	+	+	+	+	+	+	+
*fosA*		+	+	+	+	+	+	+	+
*oqxA*		ND	ND	ND	ND	ND	+	+	+
*oqxB*		ND	ND	ND	ND	ND	+	+	+
	*K. pneumoniae* LST1504-C2								
*aadA1*		ND	ND	ND	ND	ND	ND	+	+
*blaSHV*		+	+	+	+	+	+	+	+
*fosA*		+	+	+	+	+	+	+	+
*qacEΔ1*		NA	NA	NA	NA	ND	ND	+	+

^a^ See the Methods section and footnote of Table 1 for the descriptions of the assembly pipelines. ^b^ Aminoglycosides: *aadA1*, Beta-lactams: *blaACT*, *blaSHV*, *blaEC*; Phenicol: *catA1*; Fosfomycin: *fosA2*; Macrolide, tetracycline, phenicol (broad spectrum): *mdf(A)*; Tetracycline *tet(B)*, *tet(39)*; Sulphona-mide: *sul1*; Phenicol/Quinolone: *oqxA*, *oqxB*; Quaternary ammonium: *qacEΔ1*. Total genomic DNA preparations were used in this analysis since they exhibited more AMR genes, and all AMR genes detected in plasmid DNA were also detected in the total genomic preparations (see Table 3). NA: not applicable since the ResFinder database did not include biocide resistance genes. ND: gene not detected + gene detected.

## Data Availability

Bioinformatic pipeline shell script with dependencies, installation instructions, and usage instructions is available at https://github.com/iframst/HybridAMRgenotyping (accessed on 11 September 2022). Whole-genome sequences obtained from the total genomic DNA preparations on which this study is based are deposited at NCBI under the BioProject number: PRJNA624147. The plasmid sequences obtained from the plasmid DNA preparations are deposited under the BioProject number: PRJNA627408.

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
