# Peer review of "Three Distinct Annotation Platforms Differ in Detection of Antimicrobial Resistance Genes in Long-Read, Short-Read, and Hybrid Sequences Derived from Total Genomic DNA or from Purified Plasmid DNA"

_antibiotics, 2022, doi:10.3390/antibiotics11101400_

Round 1

Reviewer 1 Report

Comments to the Author

The manuscript presented by Maboni, G and coworkers is an interesting assessment of each phase of WGS to detect AMR genes in three bacterial strains, including two sequencing platforms (Illumina short-reads sequencing, Oxford Nanopore Technologies long-reads sequencing), four assembly approaches (Nanopore-only; Nanopore-polished; Illumina-only; Illumina-hybrid), two types of DNA extraction (total genomic DNA; plasmid DNA), three AMR databases (CARD; ResFinder; AMRFinder).Authors evaluated the performance of different methods for detection of AMR genes using WGS data, also made efforts to devise a best practice by correspondence of antimicrobial resistance phenotype and genotype. They emphasized the importance of standardization in AMR genes detection. Following are some my specific comments.

Major concerns:

1.     In the material and methods section, although VITEK-2 and Sensititre systems two methods used to evaluate the minimum inhibitory concentrations (MICs) of 21 antibiotics, there were several discrepant results in antimicrobial susceptibility testing. I would suggest that the authors consider using the broth microdilution (BMD)method, which was gold-standard test jointly recommended by CLSI and EUCAST. Therefore, provide a better standard to evaluate the accuracy of AMR genes detection according to phenotype results.

2.     Only three isolates were used to compare the performance of all steps for detection of AMR genes by WGS data, owing to the small number of isolates, the assessment results were subject to error and bias. So it is important to add much more isolates belonging to different species. Moreover, to evaluate the performance of the methods, accuracy and repeatability should be considered.

3.     To my knowledge, there are two main strategies to detect AMR genes from WGS data, one is assembly-based, and the other is reads-based. Besides the four assembly-based methods (Nanopore-only; Nanopore-polished; Illumina-only; Illumina-hybrid) were presented in this manuscript, I suggested the raw reads mapping strategy performed by Bowtie2 or BWA tools should be considered.

Minor concerns:

There are several grammatical errors in the manuscript. Please carefully proof read your manuscript.

1.     Line 76, ‘best’ should be ‘the best’.

2.     Line 100, ‘adapted’ should be ‘adopted’.

3.     Line 102 and Line 103, ‘hour’ should be ‘hours’

4.     Line 290, ‘of’ should be removed.

5.     Line 294, ‘a’ should be removed when used with the uncountable noun detection in your sentence.

6.     Line 491, ‘than’ should be ‘then’.

Author Response

We thank the reviewer for calling to our attention some sections requiring clarification. All requested changes specified in the minor comments were addressed in the track-changes version of the manuscript. Please, see attached a letter of response to each of the tree major points raised by the reviewer. Thank you very much.

Kind Regards.

Reviewer 2 Report

Please modify according suggested comments in attached file.

In supplementary files always illustrate E. coli as control bacteria

Author Response

We thank the reviewer for calling to our attention some sections requiring further editing and clarification. Please, see attached a letter with replies to your comments. All amendments suggested by the reviewer were addressed in the manuscript file in track changes; in addition, we left comments next to each major changes to highlight where they are placed in the modified version of the manuscript. Thank you very much.

Kind Regards

Reviewer 3 Report

This article explored different methodologies able to detect the AMR genes of pathogenic bacteria in order to obtain a more rapid diagnostic than the culture with antibiotics, especially for the slow growing bacteria. This aim is relevant even if the proposed methods like WGS are still slow and expensive to carry out. But we can hope that they will become more accessible in the future with the progresses in the microfluidic field. One of the objectives was to compare the sequencing of whole genome or plasmids for the best detection of AMR genes. The results showed clearly that the specific extraction/WGS of plasmids were not more efficient than the extraction/WGS of whole genome (chromosome + plasmid) to obtain the plasmid sequences and overall the AMR genes borne by the plasmids. Consequently it showed that plasmids are not lost along the whole genome extraction (even the small ones) and that  it was not necessary to add the plasmid  extraction step to obtain an overview of the AMR genes. Curiously the results obtained from the purified plasmids are even less complete. Several approaches of sequencing and DNA fragments assemblies were also compared which allowed to see the advantages and weaknesses of each of them. The authors also verified that the AMR genes that they detected in 3 bacterial strains were predictive of in vitro resistance. This kind of work is rare and very interesting since a lot of works present the bacterial resistomes without checking the methodology and the consistance with the phenotypic data. Then another interesting point was treated : the comparison of the 3 existing AMR databases for the detection of relevant genes. It showed that they were not equivalent and that the CARD database was less useful because it gave rise to several false positive.

This paper is well written, the discussion is clear and opens perspectives concerning the choice of the most interesting strategies to set up a robust molecular diagnostic of AMR genes.

Minor points :

L 105 showed that… ? ; L 106 : and that the K. pneumoniae… ?

Fig. 1 : is it a pulsed field electrophoresis ?

Table 1 : for the nanopore methodology, why some contigs present a  higher size than those of the corresponding plasmids ? (for example : 265,010 pb/ 153kb)

L248 : 2.6 instead of 3.6

Author Response

We thank the reviewer for calling to our attention some sections requiring clarification and further editing. Please, see attached a letter with our reply to each point raised by the reviewer. Thank you very much.

Kind Regards.

Round 2

Reviewer 1 Report

Thanks for addressing all my concerns.